# BGM-YOLO: An accurate and efficient detector for detecting plant disease

Chenghai Yu*, Junhao Xie, Fernandes Jean Adrian Tony

Department of Computer Science and Technology, Zhejiang Sci-Tech University, Zhejiang, China

* ych@zstu.edu.cn

## Abstract

Given the complexity of crop growth environments in nature, where leaf backgrounds often include soil, weeds, and other plants, along with variable lighting conditions, and considering the small size of leaf spots and the wide variety of crop diseases with significant scale differences, this paper proposes a new BGM-YOLO model structure aimed at improving accuracy and inference speed. First, the GSBottleneck module is utilized to enhance the C2f module of the YOLOv8n model, leading to the introduction of the GSC2f module, which reduces computational costs and increases inference efficiency. Next, the model incorporates a multiscale bitemporal fusion module (BFM) to increase the effectiveness and robustness of feature fusion across different levels. Finally, we developed a median-enhanced spatial and channel attention block (MECS) that combines both channel and spatial attention mechanisms, effectively improving the capture and fusion of small-scale features. The experimental results demonstrate that the BGM-YOLO model achieves a 3.9% improvement in the mean average precision (mAP) over the original model. In crop disease detection tasks, the BGM-YOLO model has higher detection accuracy and a lower false negative rate, confirming its practical value in complex application scenarios.

## 1 Introduction

Crop leaf diseases significantly impact agricultural production, making early detection and accurate identification crucial for ensuring crop health, increasing yields, and maintaining food safety [1]. Plant diseases can not only directly reduce crop yields but may also trigger other pests, further threatening the balance and stability of agricultural ecosystems [2]. With global climate change and shifts in agricultural practices, the types and transmission pathways of plant diseases have become increasingly complex, posing significant challenges to traditional monitoring and control methods. Traditional manual detection methods are time-consuming and labor-intensive, and rely heavily on the expertise and experience of the personnel involved [3]. These methods often suffer from high subjectivity and low accuracy [4], making them inadequate for large-scale agricultural production. Therefore, there is an urgent

**Data availability statement:** Quantitative data files are available from the Figshare database (via: https://doi.org/10.6084/m9.figshare.28612433).

**Funding:** The author(s) received no specific funding for this work.

**Competing interests:** The authors have declared that no competing interests exist.

and essential need to develop efficient, automated, and intelligent crop disease detection technologies [5].

With advancements in computer science and technology, machine learning algorithms have been applied in the agricultural sector. For example [6], histogram equalization is used for image preprocessing, followed by principal component analysis for feature extraction, and finally support vector machines and naive Bayes classifiers are employed to categorize rice leaf diseases. However, machine learning algorithms often face limitations due to high computational workloads during data preprocessing and feature extraction, resulting in generally poor effectiveness [7].

With the rise of convolutional neural networks and the development of residual structures, the use of deep learning algorithms for object recognition has emerged as a new research direction. Research indicates that neural networks can mimic brain mechanisms, allowing for direct self-learning from the features of the data, which endows them with strong data representation capabilities. Compared with traditional machine learning, deep learning-based object detection algorithms offer faster detection speeds, higher accuracy, and better generalizability [8], demonstrating exceptional performance in crop disease detection systems [9]. At this stage, object detection algorithms have evolved from two-stage to one-stage approaches. Among these, two-stage detection algorithms such as Faster R-CNN [10] and Mask R-CNN have been widely applied by researchers in detecting plant leaf diseases. For example, Du et al [11] used Faster R-CNN model, proposed the Pest R-CNN model to detect fall armyworms in corn fields within natural environments. This model integrates feature pyramid networks [12], attention mechanisms, deformable convolutions, and multiscale strategies. The experimental results indicate that the Pest R-CNN model significantly improves detection accuracy.

One-stage detection algorithms include SSD [13] and the YOL° [14] series, which are end-to-end methods capable of achieving high-speed real-time detection [15]. Owing the efficiency, flexibility, and strong generalizability of YOLO networks, YOLO-based disease detection algorithms have become a focal point of research [16]. For example, Liu and Wang [17] utilized the lightweight classification network MobileNetv2 as a feature extractor in MobileNetv2-YOLOv3 to increase operational efficiency. However, the numerous convolutions and bottleneck modules in the neck network result in a model with many parameters. Roy et al [18]. proposed a high-performance, fine-grained object detection framework based on YOLOv4 to address issues such as irregular shapes, multiscale targets, and similar textures in plant disease detection. Qi et al [19]. introduced a squeeze and excitation module to enhance YOLOv5's detection performance for small targets affected by tomato virus diseases. Lyu et al [20]. improved the YOLOv5s model primarily by incorporating the SE [21] attention mechanism into the backbone network, enhancing its focus on small targets. Additionally, they upgraded the CSP module in the neck network, significantly enhancing the detection accuracy for citrus leaf miners. Wang Weixing et al. improved YOLOv4-G by employing GhostNet, Ghost Modules, new feature fusion methods, and the CBAM attention mechanism.

Although previous detection methods have yielded some results in leaf disease detection, several issues remain in practical plant disease detection tasks. The complexity of crop growth environments often means that leaf backgrounds contain soil, weeds, other plants, and various lighting conditions [22]. These complex backgrounds interfere with the model's ability to extract disease features, increasing the risk of false positives and false negatives. Leaf spots are typically small and occupy a limited number of pixels in high-resolution images. Additionally, the wide variety and scale differences among crop diseases make it challenging for existing models to adapt to complex environments, leading to broad coverage of disease types [23]. While one-stage deep learning algorithms such as YOLO are fast, they often sacrifice some detection accuracy, particularly when faced with the challenges of detecting small, variable targets such as crop diseases.

To address these issues and meet the demand for rapid detection in smart agriculture, we specifically selected the smallest version of YOLOv8 as our research focus [24]. This paper uses YOLOv8n as the foundational model to propose a novel BGM-YOLO structure, focusing on enhancing the accuracy of crop disease detection while also considering detection efficiency and lightweight design. Specifically, this paper makes the following three contributions:

1. Improved C2f Module: We introduced the GSBottleneck module to enhance the C2f module, leading to the development of the GSC2f module, which effectively reduces computational load and increases inference speed.

2. Multi-Scale Bitemporal Fusion Module: The BFM module performs multi-scale fusion of features, enhancing the model's robustness and adaptability, particularly in complex backgrounds, thereby effectively reducing the risk of false positives and false negatives.

3. Median-Enhanced Spatial and Channel Attention Module: We design the MECS module in the backbone section, witch combines spatial and channel attention mechanisms to enhance the model's ability to capture and recognize multi-scale disease features.

## 2 Methods

### 2.1 Design for BGM-YOLO

Glenn Jocher proposed YOLOv8 as an optimized version of YOLOv5. This model replaces the original C3 module (which contains a CSP bottleneck with three convolutional layers) with a more efficient C2f module (a CSP bottleneck structure that includes two convolutional layers). Additionally, the number of channels in the network has been appropriately adjusted. YOLOv8 employs a decoupled head technique in its architecture, allowing classification and detection tasks to be performed separately, thus enhancing task separation. The model's loss function mechanism has also been improved by replacing traditional IOU matching with a positive-negative sample matching strategy, further enhancing training efficiency and model performance. Overall, YOLOv8's network architecture has been simplified to achieve faster detection speeds and higher accuracy [25].

This study leverages YOLOv8n as the foundational network and implements various structural optimizations to enhance its accuracy and inference efficiency in increase crop leaf diseases. First, the GSBottleneck module was introduced in the neck section, along with the proposed GSC2f module, effectively reducing the computational load and increasing the inference speed. This improvement optimizes the efficiency of feature aggregation while ensuring model accuracy. Second, the BFM module was adopted to replace traditional feature Fusion methods, significantly enhancing feature fusion effectiveness and model robustness, thereby improving the model's performance in complex backgrounds. Furthermore, multiple MECS modules were added in the backbone section, further enhancing the model's ability to capture multiscale features, particularly improving its understanding and extraction of various spatial features in complex scenes. On the basis of these improvements, the final network structure, as shown in Fig 1, reflects a comprehensive enhancement in feature extraction and fusion, effectively addressing the dual demands for accuracy and efficiency in crop disease detection tasks.

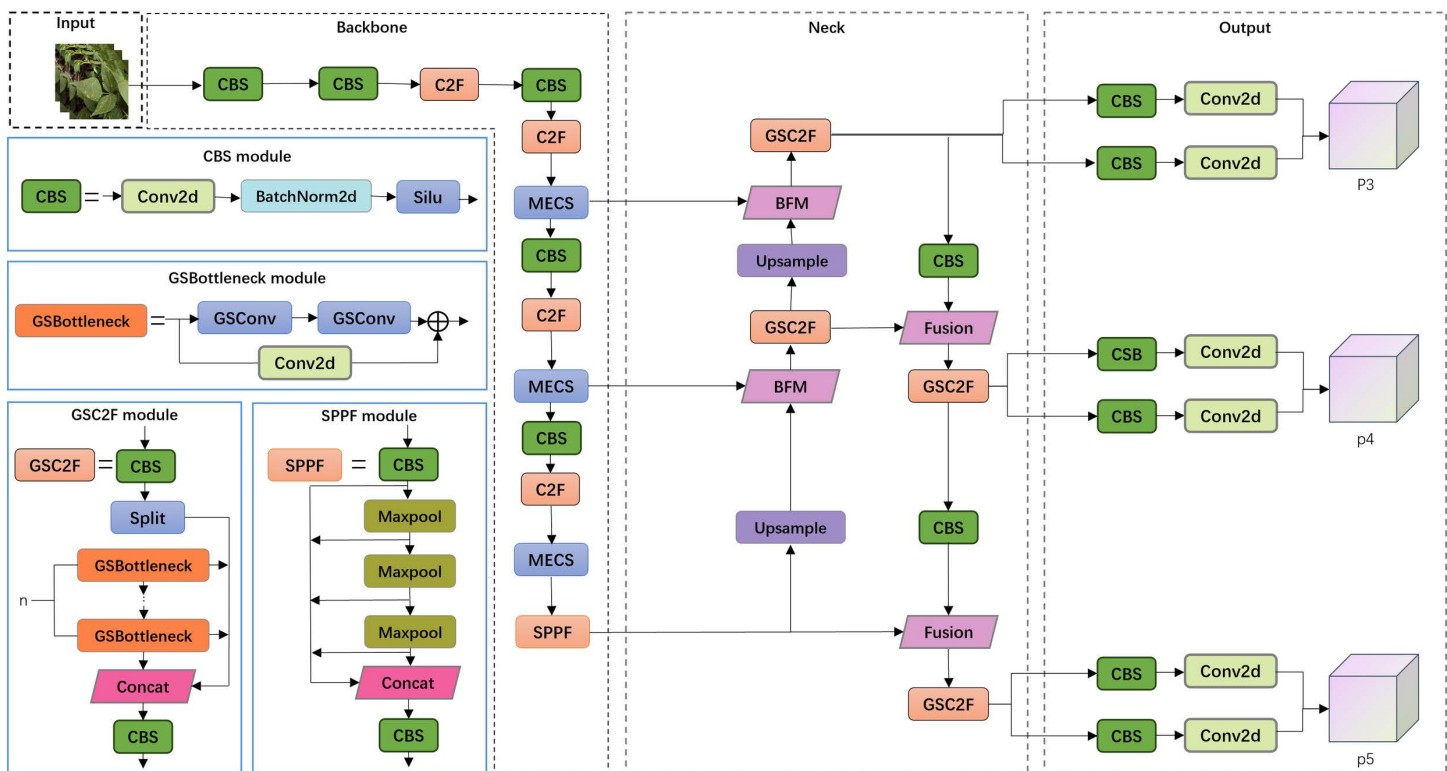

**Fig 1. The architecture of the proposed BGM-YOLO model.** The entire network is divided into four parts: the input network, backbone network, neck network, and output network (D).

## 2.2 BFM

In the traditional YOLOv8 feature fusion method (Fusion), different-sized feature maps are resized through upsampling or downsampling to match dimensions, followed by concatenation along the channel dimension to achieve feature map fusion. This process is simple and efficient, making it suitable for environments with limited computational resources. In computer vision tasks such as temporal feature analysis and change detection, a model's performance relies on its ability to effectively fuse features from different time points. However, traditional simple addition or multiplication feature fusion methods have significant limitations when handling these tasks. These methods are susceptible to noise interference, struggle to dynamically adjust the importance of features, and fail to fully leverage temporal information.

To address these issues, this paper introduces a Multi-Scale Bitemporal Fusion Module (BFM), which combines multiscale feature extraction with channel and spatial attention mechanisms to increase the effectiveness and robustness of feature fusion. This module captures feature information at different scales through multiscale convolutions and dynamically adjusts the importance of features using attention mechanisms, ensuring that key features receive higher weights during the fusion process. Within the BFM module, we introduce a more granular channel feature processing mechanism. After applying global average pooling and max pooling to the multiscale feature maps, we incorporate additional statistical information and transformation operations, such as standard deviation pooling and minimum pooling. These results are then concatenated and further processed through convolution operations to extract features, as shown in Fig 2. This approach allows better capture of complex relationships between channels, enhancing the precision and effectiveness of feature processing.

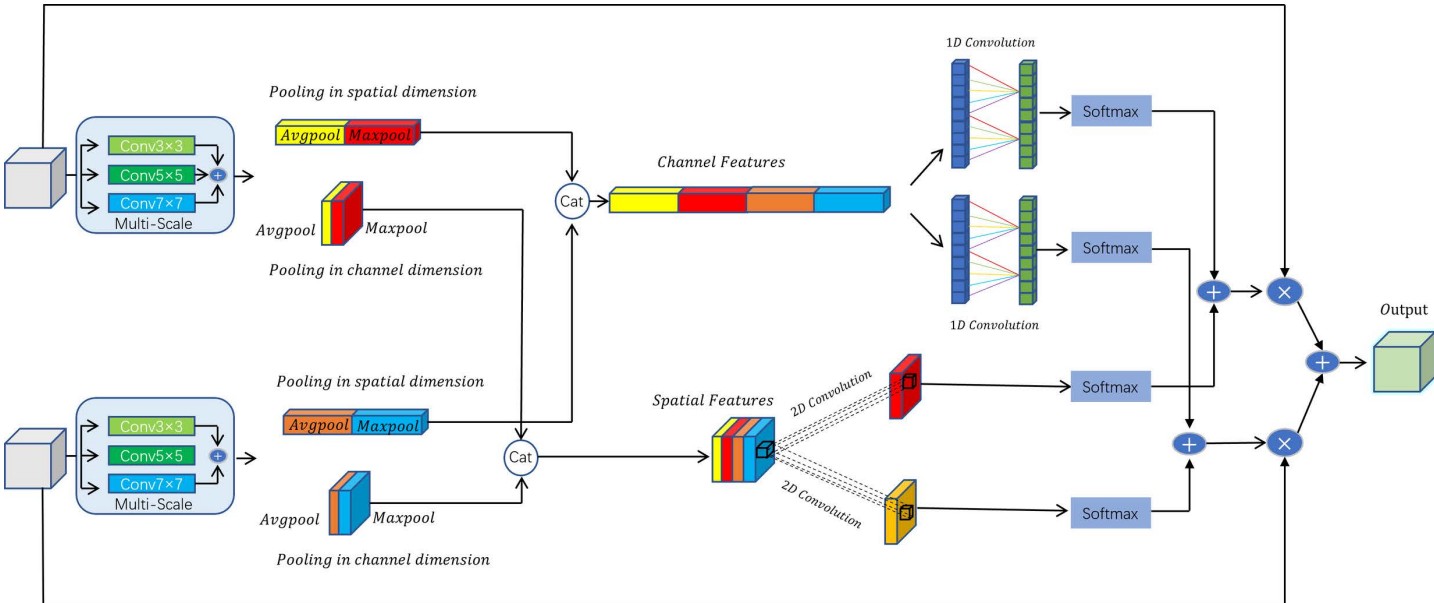

**Fig 2. BFM (Bitemporal Feature Module) flowchart: This diagram illustrates the processing pipeline of the BFM, showing the extraction and combination of channel and spatial features.** The module begins with the input feature maps, which are processed through pooling and mapping functions to enhance distinct spatial and channel features. These features are then fused and passed through convolution layers and softmax functions, yielding the final aggregated output.

This module enhances feature extraction by fusing feature maps from two different time points (T1 and T2), thereby improving the model's ability to capture and understand temporal features. First, the module applies multiscale convolution processing to the input feature maps, utilizing three different kernel sizes—3x3, 5x5, and 7x7—to extract multiscale features. This step aims to capture key information across different scales within the feature maps. Next, global pooling operations—both global average pooling and global max pooling—are performed on the multiscale feature maps generated from the convolution process. This generates global channel features and spatial features, helping the model achieve a more robust global feature representation.

Building on this, the module concatenates the pooled channel features and spatial features to create a richer feature representation. The module then calculates the attention weights for the channel features and spatial features using one-dimensional and two-dimensional convolutions, respectively. These weights are normalized using Softmax, ensuring that the sum of weights across different time points equals one. This allows the model to effectively balance the utilization of features from multiple time instances.

Finally, the module applies the computed channel and spatial weights to the feature maps of T1 and T2, performing weighted addition and multiplication to achieve feature fusion across the both channel and spatial dimensions. This fusion process effectively integrates feature information from different time points, resulting in an output feature map that is more representative and discriminative, thereby enhancing the model's ability to understand and process temporal data.

## 2.3 GSC2f

In practical applications such as plant disease detection, real-time processing of large volumes of images places high demands on the model's detection speed and accuracy. The traditional C2F module, with its multiple layers of bottleneck structures, enhances the model's feature extraction capabilities but also introduces significant computational complexity and memory overhead. This can lead to reduced inference efficiency, especially when handling simple tasks or on

resource-constrained devices, making it difficult to meet real-time detection requirements [26]. To address this issue, this paper introduces the GSBottleneck to improve the C2f module in YOLOv8, proposing the GSC2f module.

GSConv is a key component of the GSC2f module, significantly enhancing the model's overall performance. GSConv is a convolution layer that combines the advantages of Standard Convolution (SC) and Depthwise Separable Convolution (DSC), further enhancing interchannel correlations through a shuffling strategy. Its structure, as shown in Fig 3, begins with the input tensor undergoing grouped convolution through the Standard Convolution (SConv) layer, which reduces the channel dimension. Then, the output tensor performs depthwise convolution in the Depthwise Separable Convolution (DWConv) layer. Next, the outputs from SConv and DWConv are fused through a Concat layer, and the channel arrangement is further optimized in the Shuffle layer, resulting in the final output features. Compared with traditional SC and DSC, GSConv maintains and enhances high-dimensional interchannel correlations effectively while keeping computational complexity relatively low, thereby improving the model's feature extraction capabilities and inference efficiency. Its time complexity is expressed as follows:

$$T_{SC} = O(W_{out} \cdot H_{out} \cdot K_h \cdot C_1 \cdot C_2)$$
$$T_{DSC} = O(W_{out} \cdot H_{out} \cdot K_h \cdot C_1)$$
$$T_{GSConv} = O(W_{out} \cdot H_{out} \cdot K_w \cdot K_h \cdot C_1)$$

(1)

In Equation (1), $T_{SC}$, $T_{DSC}$ and $T_{GSConv}$ represent the time complexities of SC, DSC, and GSConv, respectively. $W_{out}$ and $H_{out}$ denote the width and height of the output feature map, respectively, whereas $K_w$ and $K_h$ indicate the width and height of the convolution kernels. $C_1$ represents the number of input tensor channels, and $C_2$ represents the number of output tensor channels. In the GSBottleneck network, GSConv enhances the efficiency of information fusion between channels and optimizes parameter utilization, reducing computational costs and making the model more efficient and reliable for plant disease detection.

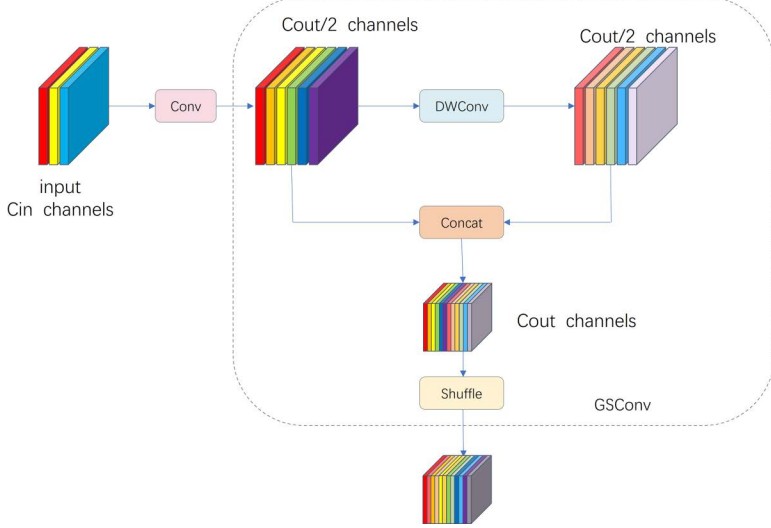

**Fig 3. GSConv Module Structure: The input is split into two groups, processed separately with depthwise convolution (DWConv) and concatenation, and then shuffled to produce the output features.**

## 2.4 MECS

The attention mechanism plays a crucial role in focusing on important areas within an image, which is beneficial for extracting small-scale occluded features related to plant diseases. However, its implementation also increases the computational workload, leading to higher costs [27]. Although YOLOv8 performs well in plant disease detection, it still faces challenges such as background interference, diversity of disease features, and small object detection. To address these issues, this paper introduces a Median-enhanced Spatial and Channel Attention Block (MECS) to effectively enhance the capability of feature extraction. Its structure, as shown in Fig 4, the MECS module combines channel and spatial attention mechanisms, enabling it to capture and fuse features across different scales.

The channel attention mechanism aggregates global statistical information from the input feature map to generate a channel attention map, which weights the channels of the input features. The specific process is as follows: the input feature map undergoes global average pooling (AvgPool), global max pooling (MaxPool), and global median pooling (MedianPool) to produce three different pooling results. Each pooling result has a size of

$$R^{C \times 1 \times 1} \tag{2}$$

Here, (C) represents the number of channels. Each pooling result is passed through a shared multilayer perceptron (MLP), which consists of two 1 x 1 convolutional layers and a ReLU activation function. The first convolutional layer reduces the feature dimension from (C) to (C/r), where (r) is the dimensionality reduction ratio, and the second convolutional layer restores the feature dimension back to (C). Finally, the Sigmoid activation function is applied to compress the output values into the range of [0, 1], resulting in three attention maps. The three attention maps from the pooling results are summed elementwise to obtain the final channel attention map. The channel attention map is then elementwise multiplied with the original input feature map to produce the weighted feature map. The formula is as follows:

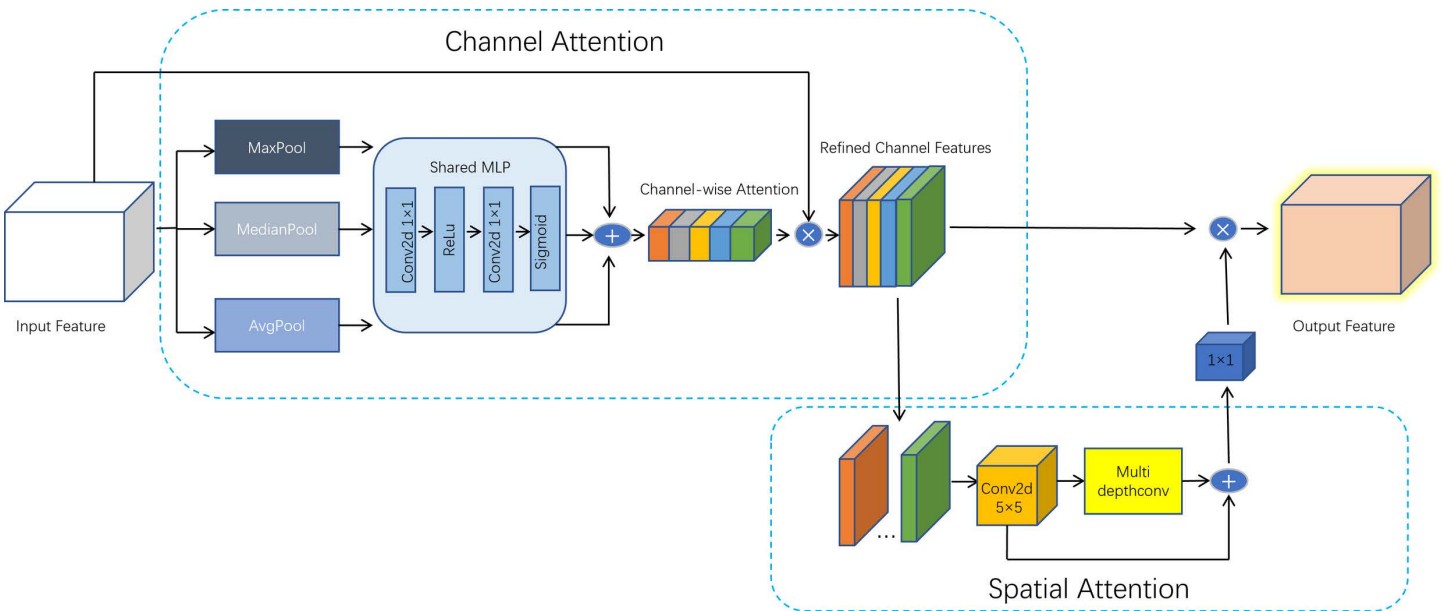

**Fig 4. GSConv Module Structure: The input is a split MECS Module Structure: This module combines channel and spatial attention mechanisms.** Channel attention uses MaxPool, MedianPool, and AvgPool operations, which are processed through a shared MLP to generate refined channel features. Spatial attention applies convolution and multidepth convolutions to capture spatial information, enhancing feature representation.

$$F_c = \sigma(\text{MLP}(\text{AvgPool}(F))) + \sigma(\text{MLP}(\text{MaxPool}(F))) + \sigma(\text{MLP}(\text{MedianPool}(F))) \tag{3}$$

$$F' = F_c \odot F \tag{4}$$

Here, $\sigma$ denotes the Sigmoid function, and $\odot$ represents elementwise multiplication.

The spatial attention mechanism captures the spatial relationships of the input feature map through multiscale depthwise convolutions, generating a spatial attention map. The process is as follows: the input feature map is first passed through a $5 \times 5$ depthwise convolutional layer to extract basic features. The output size of this convolutional layer is the same as that of the input. The output feature map from the initial convolutional layer is then passed through multiple depthwise convolutional layers of varying sizes, such as $1 \times 11$, $1 \times 7$, and others, to further extract multi-scale features. The outputs from all depthwise convolutional layers are summed elementwise to obtain the fused feature map. The fused feature map is then passed through a $1 \times 1$ convolutional layer to generate the final spatial attention map. This attention map is element-wise multiplied with the channel-weighted feature map to produce the final output feature map. The formula is as follows:

$$F_s = \sum_{i=1}^{n} D_i(F') \tag{5}$$

$$F'' = Conv\text{1x1}(F_s) \odot F' \tag{6}$$

Here, $D_i$ indicates the depthwise convolution operations of different sizes, $n$ represents the number of depthwise convolutions, and $Conv1 \times 1$ indicates the convolution operation of $1 \times 1$.

## 3 Materials and results

### 3.1 Dataset

In this study, we address the common challenges in plant disease detection tasks, such as variations in lighting, complex backgrounds, and difficulties in d distinguishing between different disease features. We utilized a plant disease object detection dataset from the Roboflow open-source platform, which underwent the systematic preprocessing. This dataset aims to increase the automation level of agricultural disease detection and covers a wide range of plant disease categories, including ALS, angular leaf spot, anthracnose, rust, gray mold, leaf spot, powdery mildew, and other major diseases, as well as pests such as spider mites, comprehensively covering common plant disease types in agricultural production.

The dataset consists of 5,841 high-quality images, divided into 4,088 for training, 1,168 for validation, and 585 for testing following a 7:2:1 ratio, ensuring a balanced distribution for model training and evaluation. It includes images from three different plant species (Beans, Strawberries, and Tomatos), with a total of 12 plant disease categories along with their corresponding healthy classes. The most represented category is Beans_Rust with 800 images, while the least represented is Strawberry_Anthracnose_Fruit_Rot with 300 images, indicating a certain level of class imbalance. On average, each image contains 2 annotated objects, and the bounding box sizes range from 200 to over 8,000 pixels, covering a variety of lesion scales, from early-stage infections to widespread diseases. This dataset provides a diverse and comprehensive representation of plant diseases under different conditions, as shown in Fig 5, ensuring robust model training while addressing real-world complexities in agricultural disease detection.

To enhance the model's generalizability and robustness in complex scenarios, the dataset underwent strict preprocessing before use. The preprocessing steps included standardizing the image size to ensure consistent resolution, thereby optimizing the model's processing efficiency; adjusting brightness and contrast to address differences in image quality

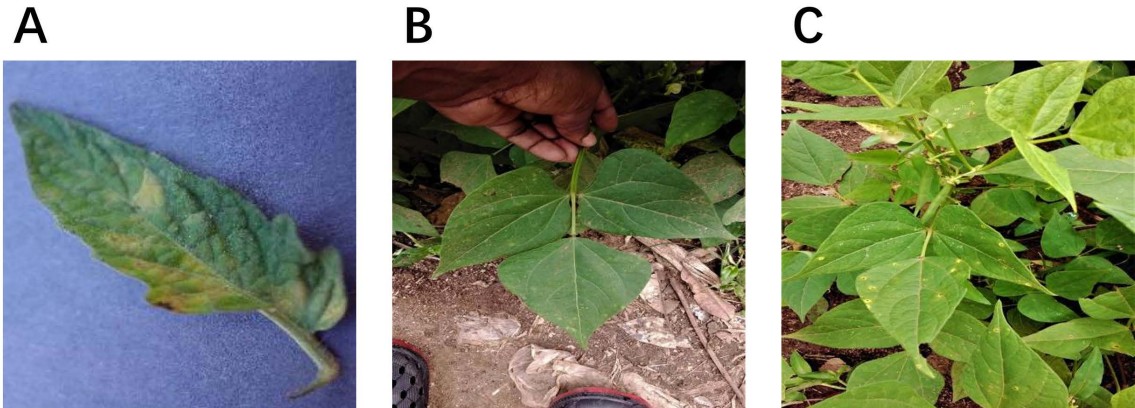

**Fig 5. Samples of datasets, where (A) is a diseased apple leaf with a normal background, (B) is a diseased apple leaf in a real environment, and (C) is in an intensive situation where samples of apple leaves have multiple diseases collected in this paper.**

under various lighting conditions; and employing data augmentation techniques such as random cropping, rotation, and horizontal flipping to enrich the diversity of the training data, thereby improving the model's adaptability to disease features from different angles and complex backgrounds.

These preprocessing measures not only ensure the quality and consistency of the dataset but also enhance the diversity of samples, allowing the model to demonstrate greater robustness and accuracy in complex environments. After preprocessing, the model was better able to learn and recognize plant disease features, significantly improving precision and efficiency in plant disease detection tasks. This provided a solid data foundation for subsequent model optimization and application.

### 3.2 Experimental environment and hyperparameters

The experiments were conducted on a server with an Intel(R) Xeon(R) CPU E5-2696 v3 processor, an RTX 3090 GPU (24GB), and 64GB RAM, running Ubuntu as the operating system. The model was implemented using Python 3.8 and trained with CUDA 12.1 for GPU acceleration. The network was trained for 300 epochs with a batch size of 16, using the SGD optimizer with a learning rate of 0.0001. The input image size was 640×640 pixels. Unlike many previous studies, no pretrained models were used, and the network was trained from scratch to learn domain-specific features directly from the plant disease dataset.

### 3.3 Evaluation indicators

In the field of object detection, the most commonly used evaluation indicators are mAP50, mAP50:95, GFLOPs, Params, FPS and P (precision):

$$P = \frac{TP}{TP + FP} \times 100\%$$

(7)

$$R = \frac{TP}{TP + FN} \times 100\%$$

(8)

$$mAP = \frac{\sum\limits_{n \in N} AP(n)}{N}$$

(9)

$$FLOPs = 2HW(C_{in}K^2 + 1)C_{out} + (2I - 1)O \qquad (10)$$

In this study, the metrics used include $W$ and $H$ for the height and width of the input feature map, $C_{in}$ for the number of input channels, $K$ for the width of the convolutional kernel, $C_{out}$ for the number of output channels, $I$ for the input dimensions, and $O$ for the output dimensions. TP refers to the number of true positive samples detected, FP indicates the number of false positives, FN denotes the number of false negatives, and mAP represents the average precision across all categories.

Specifically, mAP50 and mAP50:95 are used to evaluate the accuracy of the model's object detection, with higher values indicating greater accuracy. The mAP50 metric calculates mean Average Precision at an Intersection over Union (IoU) threshold of 0.5, measuring how well predicted bounding boxes overlap with ground truth annotations. A prediction is considered correct when it shares at least 50% overlap with the actual object. The mAP50:95 provides a more comprehensive evaluation by averaging AP values across multiple IoU thresholds from 0.5 to 0.95 (in 0.05 increments), emphasizing the model's precision in accurately localizing objects and delineating their boundaries. This is particularly important for plant disease detection, where precise localization of affected areas is essential.

The real-time performance of the model is measured in frames per second (FPS), which indicates the number of images the model can detect per second. GFLOPs and the number of parameters are used to assess the model's computational efficiency and suitability for deployment in specific environments. As training progresses, the BGM-YOLO model's loss curve converges to a lower level, and the prediction accuracy stabilizes in the later stages of training, as shown in Fig 6.

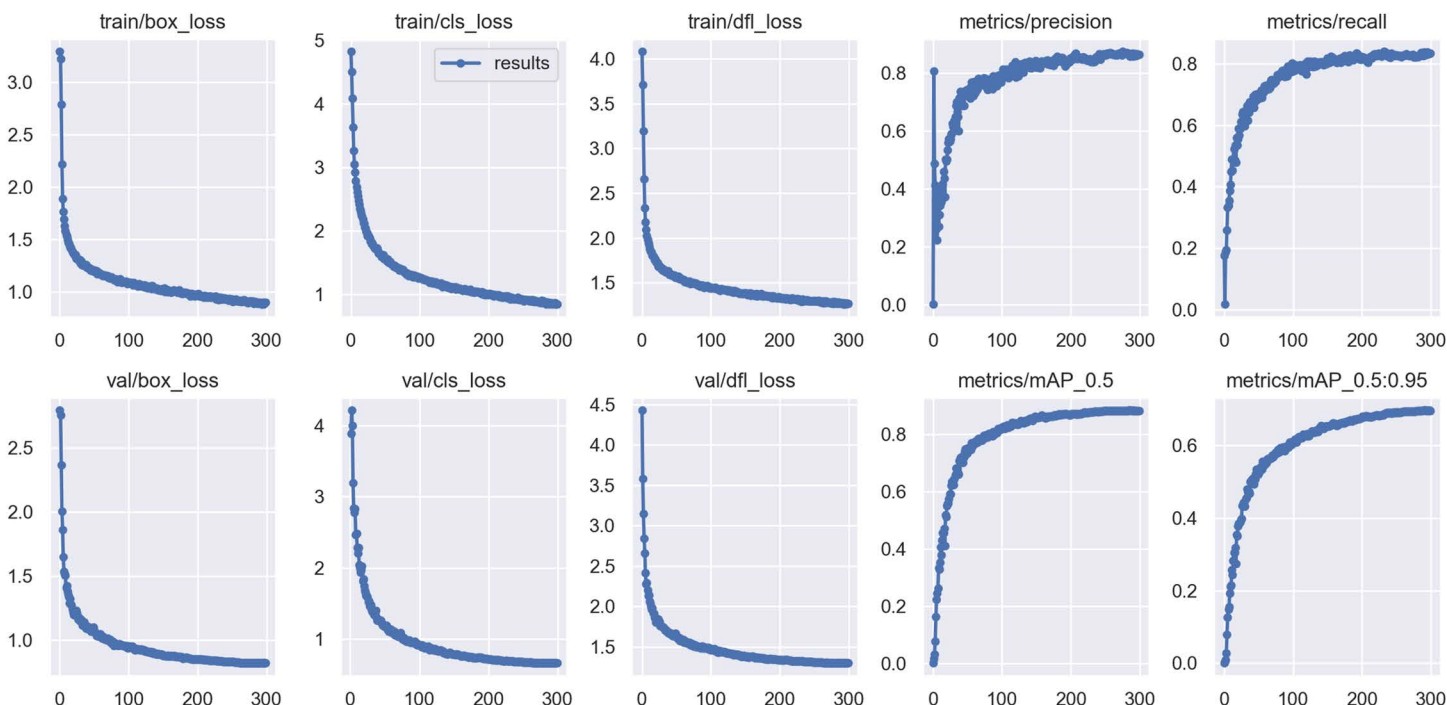

**Fig 6. Training and validation loss curves along with performance metrics (precision, recall, mAP@0.5, and mAP@0.5:0.95) across epochs for the BGM-YOLO model.** The graphs indicate the model's convergence and improvements in accuracy metrics over time.

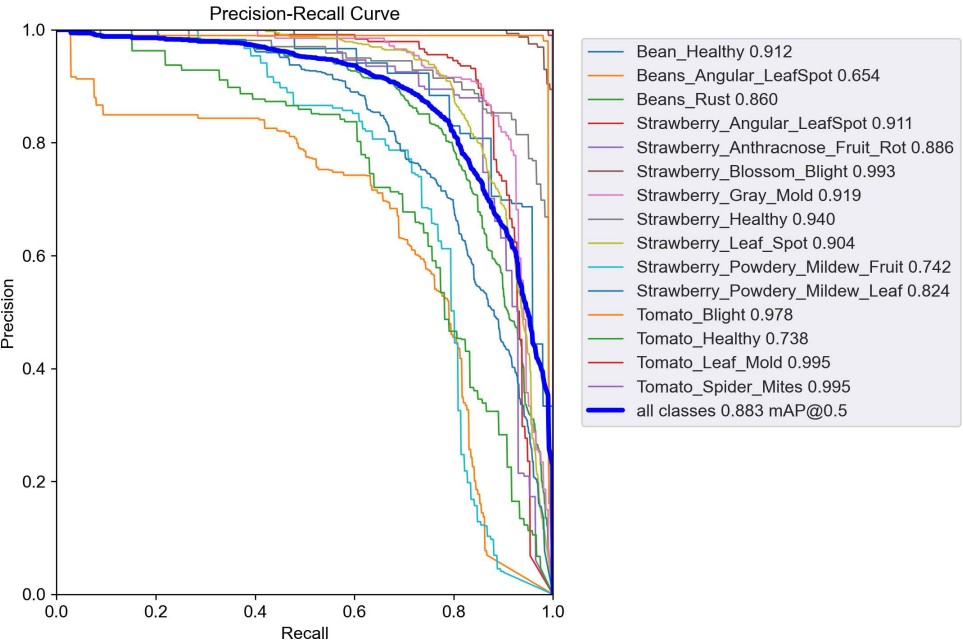

**Fig 7. Precision-Recall curve for various plant disease categories, illustrating the detection performance of the BGM-YOLO model across different disease types.** Each color represents a specific disease category, with mAP@0.5 values annotated on the right. The bold blue line indicates the overall performance across all categories, achieving an average mAP@0.5 of 0.883.

The experiment also generated Precision-Recall curves for each category, as shown in Fig 7. BGM-YOLO successfully identified the 15 defined categories, achieving an average recognition accuracy of 99.5% for "Tomato Leaf Mold" and "Tomato Spider Mites." The overall average recognition accuracy across all categories reached 88.3

To better understand the model's classification performance, we present the confusion matrix for the test dataset in Fig 8. It highlights both correct predictions and misclassifications across different disease categories. The strong diagonal values indicate high accuracy for most classes, but some misclassifications exist, particularly between visually similar diseases such as Beans_Angular_LeafSpot and Beans_Rust, or Strawberry_Powdery_Mildew_Leaf and Strawberry_Powdery_Mildew_Fruit. These errors suggest that certain disease features are difficult to distinguish, leading to model confusion.

To further assess detection performance across plant categories, we evaluated Precision (P), Recall (R), mAP50, and mAP50:95 for each class, as shown in Table 1. The overall mAP50 reached 0.883, while mAP50:95 was 0.697, demonstrating strong detection capability. Among healthy classes, Bean_Healthy and Strawberry_Healthy achieved high mAP50 scores of 0.912 and 0.939, respectively, indicating that the model effectively differentiates healthy from diseased plants. However, Tomato_Healthy showed lower accuracy, with an mAP50 of 0.737 and an mAP50:95 of 0.411, likely due to sample imbalance or the visual similarity between healthy and diseased leaves.

For disease detection, classes such as Strawberry_Anthracnose_Fruit_Rot, Strawberry_Gray_Mold, Tomato_Blight, and Tomato_Leaf_Mold had mAP50 values above 0.90, demonstrating strong performance. However, Beans_Angular_LeafSpot, Strawberry_Powdery_Mildew_Fruit, and Strawberry_Powdery_Mildew_Leaf showed relatively lower mAP50 values of 0.654, 0.742, and 0.824, respectively, suggesting room for improvement.

As shown in Fig 8, confusion matrix analysis confirms that diseases with similar visual features are more prone to misclassification. Addressing this challenge could involve increasing data diversity through augmentation techniques such as color transformations and contrast enhancement. Additionally, integrating attention mechanisms could refine feature extraction, enhancing classification accuracy for visually similar disease categories.

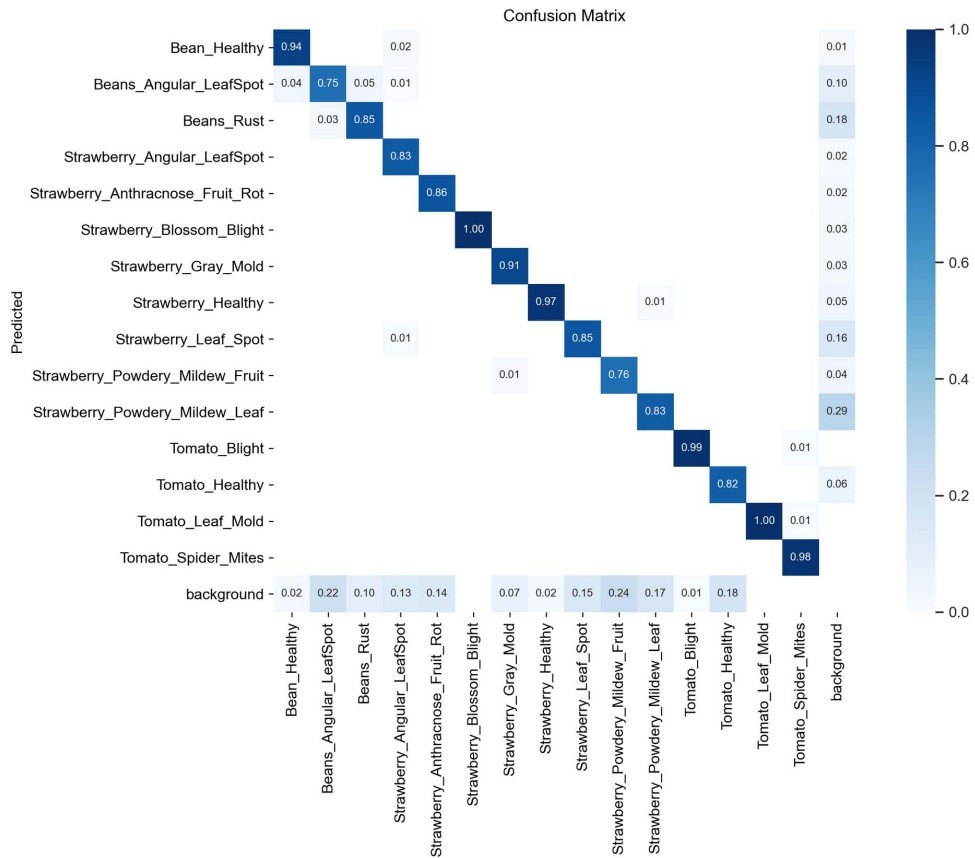

**Fig 8. Comparison of model performance with different training data volumes.** The detection accuracy (mAP50 and mAP50:95) of both the baseline and proposed models increases as the training data percentage grows. The proposed model consistently outperforms the baseline, especially when the training sample size exceeds 60%, demonstrating its superior ability to learn disease features and detect small objects.

## 3.4 Comparative experimental results and analysis of different attention mechanisms

To evaluate the effectiveness of the Median-Enhanced Spatial and Channel Attention Mechanism (MECS) in plant disease detection tasks, this study designed a series of rigorous comparative experiments aimed at analysing its performance relative to other mainstream attention mechanisms. We selected several widely used attention mechanisms in the field of object detection for comparison, including the Squeeze-and-Excitation (SE) module, Convolutional Block Attention Module (CBAM), and Efficient Channel Attention (ECA) module. These mechanisms have demonstrated excellent feature extraction capabilities across various tasks.

The experimental results indicate that the Median-Enhanced Spatial and Channel Attention Mechanism (MECS) outperformed all other models, as shown in Table 2, achieving an mAP50 of 0.854, which is significantly higher than those of the other attention mechanisms. This finding demonstrate the superior ability of the MECS to capture plant disease feature information accurately. Although the introduction of MECS increased the number of GFLOPs to 9.2, slightly higher than both the baseline model and other attention mechanism models, this increase in computational cost is reasonable and acceptable given the performance gains. In contrast, while the CBAM attention mechanism also increased the computational complexity (to 8.9 GFLOPs), its mAP50 was only 0.847, failing to surpass MECS's performance. Therefore, MECS excels at balancing detection accuracy and computational efficiency, significantly enhancing the model's accuracy while only slightly increasing computational costs.

**Table 1. The detection performance different categories of diseased.**

| Class | Instances | P | R | mAP50 | mAP50:95 |
|---|---|---|---|---|---|
| all | 3480 | 0.866 | 0.828 | 0.883 | 0.696 |
| Bean_Healthy | 98 | 0.788 | 0.875 | 0.912 | 0.798 |
| Beans_Angular_LeafSpot | 277 | 0.681 | 0.671 | 0.654 | 0.407 |
| Beans_Rust | 463 | 0.812 | 0.782 | 0.86 | 0.61 |
| Strawberry_Angular_LeafSpot | 193 | 0.956 | 0.746 | 0.911 | 0.679 |
| Strawberry_Anthracnose_Fruit_Rot | 85 | 0.874 | 0.815 | 0.886 | 0.62 |
| Strawberry_Blossom_Blight | 161 | 0.929 | 0.988 | 0.993 | 0.881 |
| Strawberry_Gray_Mold | 200 | 0.896 | 0.85 | 0.914 | 0.59 |
| Strawberry_Healthy | 120 | 0.819 | 0.925 | 0.939 | 0.751 |
| Strawberry_Leaf_Spot | 779 | 0.9 | 0.796 | 0.905 | 0.801 |
| Strawberry_Powdery_Mildew_Fruit | 151 | 0.804 | 0.653 | 0.742 | 0.575 |
| Strawberry_Powdery_Mildew_Leaf | 534 | 0.763 | 0.728 | 0.824 | 0.674 |
| Tomato_Blight | 100 | 0.974 | 0.99 | 0.978 | 0.835 |
| Tomato_Healthy | 119 | 0.801 | 0.605 | 0.737 | 0.411 |
| Tomato_Leaf_Mold | 101 | 0.994 | 0.99 | 0.995 | 0.861 |
| Tomato_Spider_Mites | 99 | 0.995 | 0.99 | 0.995 | 0.956 |

**Table 2. Hyperparameter settings of network training.**

| Model | mAP50 | GFLOPs | Params |
|---|---|---|---|
| YOLOv8n | 0.844 | 8.6 | 3.0 M |
| YOLOv8n +CBAM | 0.847 | 8.9 | 3.2 M |
| YOLOv8n+SE | 0.850 | 8.7 | 3.1 M |
| YOLOv8n+EMA | 0.848 | 8.6 | 3.1 M |
| YOLOv8n +MECS | **0.854** | **9.2** | **3.3M** |

In summary, MECS demonstrates superior feature extraction capabilities while maintaining high computational efficiency, particularly in complex plant disease detection tasks, where it effectively enhances the model's detection accuracy. Thus, selecting MECS as the attention mechanism is based on its overall performance, which exceeds that of other mainstream mechanisms across various metrics.

## 3.5 Comparison and analysis

To comprehensively and objectively evaluate the performance of the BGM-YOLO model, a comparison was made with currently mainstream lightweight object detection algorithms, as shown in Table 3.

As shown in Table 3, the proposed BGM-YOLO achieves optimal results compared with other models with similar parameter counts, with improvements of 3.9% and 5.8% in term of mAP@0.5% and mAP@0.5:0.95% over YOLOv8n, 3.7% and 5.2% over YOLOv9t, and 5.1% and 4.9% over YOLOv10n. Although the number of GFLOPs is greater than that of YOLOv8n, it still meets the lightweight requirements for mobile devices. During the training of YOLOv10, the model combines one-to-many and one-to-one strategies for efficient end-to-end deployment, eliminating the reliance on Non-Maximum Suppression (NMS). However, despite the provision of auxiliary gradient information through the one-to-many detection head to optimize model training, YOLOv10 still faces certain limitations in complex and dense detection scenarios. In contrast, BGM-YOLO demonstrates superior accuracy in handling such complex tasks, indicating a decline in the detection performance of YOLOv10 for these types of tasks.

**Table 3. Comparative experiments.**

| Model | mAP50/% | mAP50:95/% | Params/M | GFLOPs |
|---|---|---|---|---|
| YOLOv5n | 81.5 | 59.5 | 2.5 | 7.5 |
| YOLOv5s | 86.7 | 65.2 | 4.6 | 12.4 |
| YOLOv8n | 84.4 | 63.8 | 3.0 | 8.6 |
| YOLOv9t | 84.6 | 64.4 | 2.0 | 5.8 |
| YOLOv10n | 83.2 | 64.7 | 2.3 | 6.7 |
| Ours | 88.3 | **69.6** | **4.1** | **10.9** |

## 3.6 Ablation experiment and analysis

To better validate the performance improvements of each modified module and their combinations on the original model, ablation experiments were designed, and the results are shown in Table 4. The experimental results indicate that each module introduced in this study contributes to varying degrees of improvement in model accuracy. Notably, the MECS and BFM modules significantly enhance model performance; when used together, they improve mAP@0.5 and mAP@0.5:0.95 by 3.3% and 5.1%, respectively. Furthermore, the GSC2F module significantly optimizes the model's inference efficiency, effectively reducing both computation and parameter counts while maintaining high accuracy, leading to a notable increase in FPS. Compared with the original model, the combination of MECS and GSC2F improves the mAP@0.5 and mAP@0.5:0.95 by 1.5% and 2.3%, respectively, while also optimizing the GFLOPs and parameter counts. In contrast, the BFM and GSC2F combination focuses primarily on enhancing overall model accuracy, achieving improvements of 3.1% and 5% in mAP@0.5 and mAP@0.5:0.95, respectively, although it results in a decrease in inference speed. Ultimately, when all three modules are used together, the model achieves improvements of 3.9% and 5.8% in terms of mAP@0.5 and mAP@0.5:0.95, respectively, while maintaining a high inference speed of 49 FPS, demonstrating the significant advantages of their combination in enhancing detection accuracy and optimizing computational efficiency. Overall, the results of the ablation experiments show that each module and its combination effectively enhance the model's detection performance, demonstrating their respective advantages in various application scenarios.

To investigate the impact of training data volume on model accuracy, we trained the model using 20%, 40%, 60%, 80%, and 100% of the training dataset and evaluated its performance on the test set. As shown in Fig 9, the detection accuracy (mAP50 and mAP50:95) steadily improves as the amount of training data increases. Notably, when the data volume increases from 60% to 100%, the accuracy gain tends to stabilize, indicating a diminishing marginal improvement in model performance.

Furthermore, we compared our method with baseline models YOLOv8n under the same data conditions. The results show that when the training sample size exceeds 60%, our method begins to outperform the baseline models, achieving

**Table 4. Analysis of algorithm improvement ablation experiments.**

| MECS | BFM | GSC2F | mAP50/% | mAP50:95/% | GFLOPs | Params/M | FPS |
|---|---|---|---|---|---|---|---|
| ✗ | ✗ | ✗ | 84.4 | 63.8 | 8.6 | 3.0 | 64 |
| ✓ | ✗ | ✗ | 85.3 | 64.7 | 9.2 | 3.3 | 58 |
| ✗ | ✓ | ✗ | 86.1 | 67.7 | 10.1 | 4.0 | 48 |
| ✓ | ✓ | ✗ | 87.7 | 68.9 | 11.0 | 4.3 | 45 |
| ✓ | ✗ | ✓ | 85.9 | 66.1 | 8.3 | 2.9 | 67 |
| ✗ | ✓ | ✓ | 87.2 | 68.6 | 10.2 | 3.9 | 50 |
| ✓ | ✓ | ✓ | **88.3** | **69.6** | **10.9** | **4.1** | **49** |

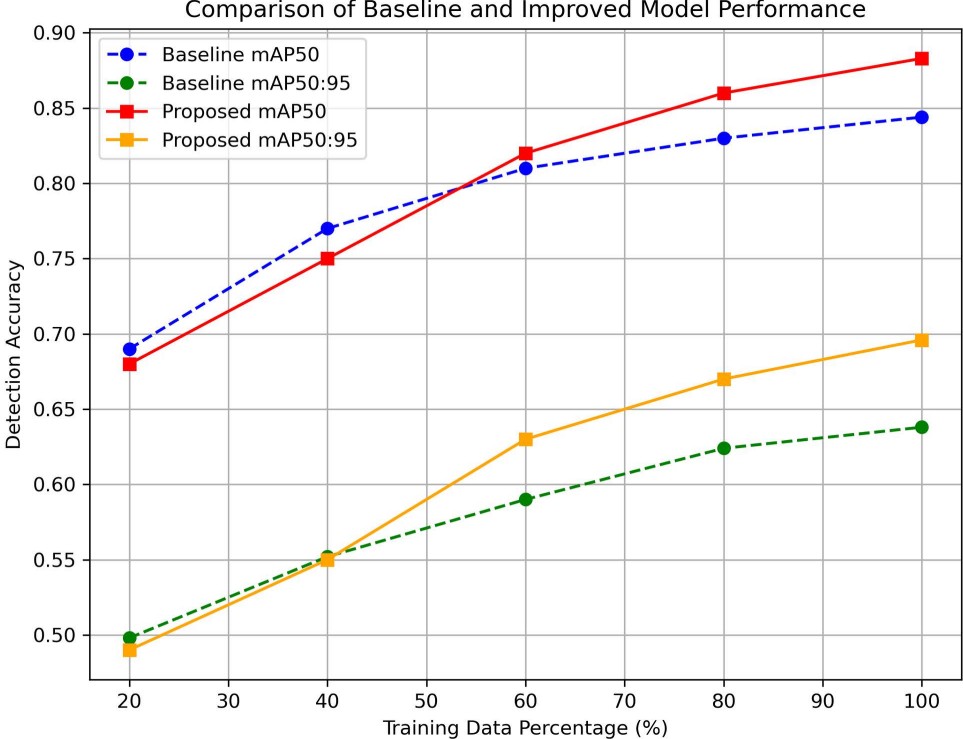

Fig 9. Comparison of model performance with different training data volumes. The detection accuracy (mAP50 and mAP50:95) of both the baseline and proposed models increases as the training data percentage grows. The proposed model consistently outperforms the baseline, especially when the training sample size exceeds 60%, demonstrating its superior ability to learn disease features and detect small objects.

superior detection accuracy. This demonstrates that our improved network can more effectively learn disease features and enhance small-object detection capabilities when provided with sufficient training data.

### 3.7 Generalization performance on an external dataset

To further verify the generalization ability of the model, we conducted experiments using the tomato leaf disease detection dataset provided by Md Faruk Alam. This dataset is annotated in YOLOv5 PyTorch format and includes seven categories: Bacterial Spot(BS), Early Blight(EB), Healthy(H), Late Blight(LB), Leaf Mold(LM), Target Spot(TS), and Black Spot(BSp), covering various types of tomato leaf diseases.

The dataset contains a total of 700 images, which is significantly smaller than our original dataset of 5,800 images. Additionally, there are some differences in disease categories between the two datasets, making it suitable for evaluating the model's adaptability to new data. However, due to the small dataset size, directly using it for testing may lead to evaluation results being heavily influenced by data distribution and even potential overfitting. To address this, we applied data augmentation techniques, including random rotation, flipping, color jittering, and Gaussian noise, expanding the dataset to 3,000 images. These augmentations help better simulate real-world scenarios and improve the reliability of the generalization evaluation.

Table 5 presents the detection performance of YOLOv8n and the improved model (Ours) on the tomato leaf disease dataset. The experiment compares the Average Precision (AP) for different disease categories and the mean Average Precision at IoU = 0.5 (mAP@0.5) to evaluate the model's generalization ability.

The results indicate that the improved model achieves higher AP in most categories, particularly in Bacterial Spot (BS) (from 68% to 74.6%), Early Blight (EB) (85.7%→94.3%), and Target Spot (TS) (62.7%→73.9%), demonstrating

**Table 5. Comparison of generalization performance on the tomato leaf disease dataset.**

| Model | AP/% | | | | | | | mAP@0.5/% |
|---|---|---|---|---|---|---|---|---|
| | BS | EB | H | LB | LM | TS | BSp | |
| YOLOv8n | 68 | 85.7 | 99.5 | 89.8 | 61.3 | 62.7 | 50 | 79.3 |
| Ours | 74.6 | 94.3 | 99.5 | 88 | 53.6 | 73.9 | 51.1 | 83.5 |

enhanced recognition capability for these disease types. However, a slight decrease in AP is observed in Late Blight (LB) (89.8%→88%) and Leaf Mold (LM) (61.3%→53.6%), which may be attributed to the data augmentation methods altering intra-class features, affecting the model's discrimination ability.

Despite the decrease in AP for some categories, the overall mAP@0.5 remains at 79.3%, comparable to YOLOv8n, indicating that the improved model maintains generalization capability while enhancing detection performance for specific disease categories. This suggests that, without additional training data, the optimization methods effectively improve detection for certain disease types while preserving overall model stability. Future improvements could focus on refining data augmentation strategies or incorporating more advanced feature extraction mechanisms to further enhance generalization performance.

### 3.8 Visual analysis

To provide a clearer comparison of the detection performance between the proposed algorithm and the YOLOv8n algorithm, both algorithms were tested on the same dataset. Representative images were selected for visual analysis, with the comparative results illustrated in the Fig 10 The analysis focuses on complex scenes containing strawberry fruits and leaves for a comparison of detection results.

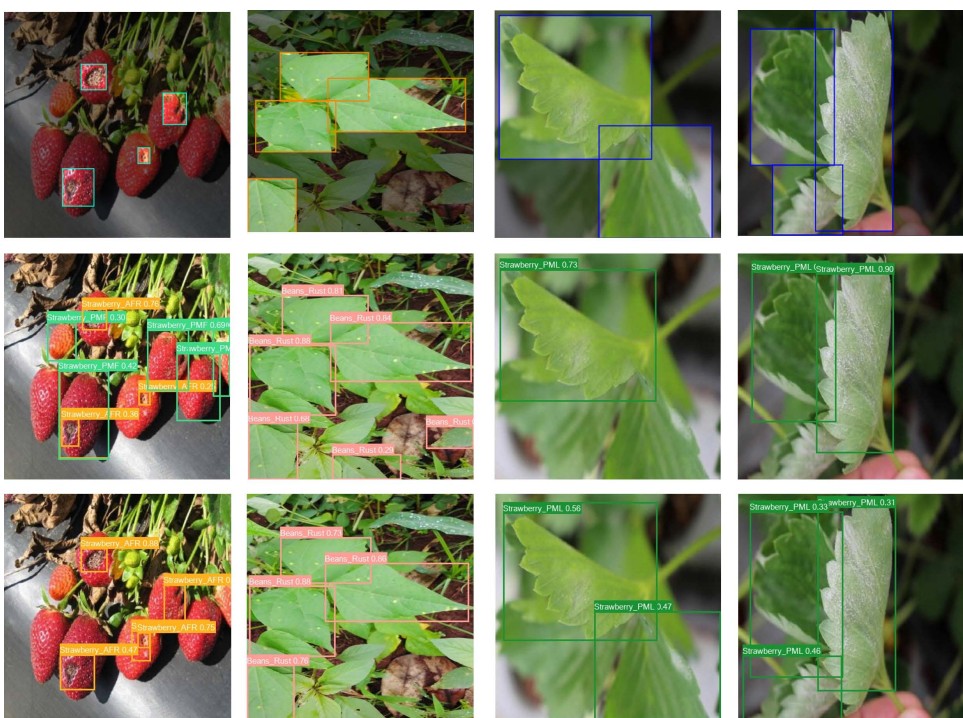

**Fig 10. Detection comparison in complex scenes: the top row shows the original annotations, the middle row shows the YOLOv8n results, and the bottom row shows the BGM-YOLO results.** BGM-YOLO performs better in detecting small lesions, handling complex backgrounds, and distinguishing targets, reducing missed detections.

The top section of the figure displays the original annotated image, the middle section shows the detection results from YOLOv8n, and the bottom section presents the detection results from the improved network (BGM-YOLO).

Comparative analysis of the visualization results reveals that the improved algorithm can more accurately capture subtle differences in the targets, showing significantly enhanced resistance to interference, particularly in complex backgrounds. The improved algorithm demonstrates notable enhancements in detecting small lesions and details, effectively reducing instances of missed detections and achieving greater accuracy in classifying diseased leaves and strawberry fruits. Compared with YOLOv8n, BGM-YOLO excels at distinguishing targets from the background, even under challenging conditions such as complex leaf textures or poor lighting (e.g., dim or bright environments), minimizing the risk of false detections.

Furthermore, the BGM-YOLO network maintains strong robustness even in complex backgrounds and scenarios with target occlusion, accurately perceiving the overall edges and shapes of the targets while effectively reducing both false positives and missed detections. Compared with the original YOLOv8n, the improved algorithm performs exceptionally well in detecting fruits and leaves, demonstrating enhanced detection efficacy and stability in complex scenes.

In summary, the improved BGM-YOLO method demonstrates outstanding detection performance in handling complex scenes, subtle differences, background interference, and varying lighting conditions. This method exhibits enhanced robustness and accuracy, effectively improving the overall performance of strawberry disease detection

## 4 Conclusion

This paper presents an improved YOLOv8 object detection model aimed at enhancing the accuracy and efficiency of plant disease detection. Compared with previous models, this study introduces several innovations and optimizations. First, the incorporation of the Median-Enhanced Spatial and Channel Attention Mechanism (MECS) significantly enhances the model's ability to detect small targets and its robustness in complex environments. Second, the combination of multiscale convolution operations strengthens the model's feature extraction capabilities, ensuring stable performance across different scales. Furthermore, the introduction of the GSBottleneck to improve the C2f module increases the model's detection accuracy, particularly in addressing the diversity and complexity of plant disease targets. The experimental results indicate that the proposed BGM-YOLOv8 model outperforms existing mainstream attention mechanism models in several performance metrics, including mAP and GFLOPs, demonstrating excellent detection effectiveness and computational efficiency. However, despite its outstanding performance in plant disease detection, the BGM-YOLOv8 model has serveral limitations. First, its high computational complexity impacts its applicability to resource-constrained devices. Second, the model's adaptability in complex scenarios and its generalization performance need improvement. Additionally, the model lacks real-time capabilities, and fails to meet low-latency requirements. Moreover, the limited diversity of the dataset may lead to inconsistent performance in real-world applications, indicating a need for dataset expansion. Finally, further exploration is needed to achieve model lightweighting for effective deployment on edge devices. Future research will focus on addressing these issues by employing techniques such as pruning, quantization, and knowledge distillation to reduce the model parameters and computational load, thereby increasing the deployment capabilities on edge devices. Additionally, expanding the dataset will improve the model's generalization performance and optimize its adaptability in complex scenarios, promoting the widespread application of this model in the agricultural sector.

## Author contributions

**Formal analysis:** Fernandes Jean Adrian Tony.

**Investigation:** Chenghai Yu.

**Methodology:** Junhao Xie.

**Project administration:** Chenghai Yu.

**Supervision:** Fernandes Jean Adrian Tony.

**Writing – original draft:** Junhao Xie.

**Writing – review & editing:** Junhao Xie.

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
