## [Decision Letter · Decision Letter 0]

5 Mar 2025

Dear Dr. Xie,

Thank you for submitting your manuscript to PLOS ONE. After careful consideration, we feel that it has merit but does not fully meet PLOS ONE’s publication criteria as it currently stands. Therefore, we invite you to submit a revised version of the manuscript that addresses the points raised during the review process.

We look forward to receiving your revised manuscript.

Kind regards,

Hirenkumar Kantilal Mewada

Academic Editor

PLOS ONE

Journal Requirements:

4. Please ensure that you refer to Figure 2, 4, 5, 6 and 8 in your text as, if accepted, production will need this reference to link the reader to the figure.

5. We note you have included a table to which you do not refer in the text of your manuscript. Please ensure that you refer to Table 1, 2, 3, and 5 in your text; if accepted, production will need this reference to link the reader to the Table.

Reviewers' comments:

Reviewer's Responses to Questions

**Comments to the Author**

1. Is the manuscript technically sound, and do the data support the conclusions?

Reviewer #1: Yes

Reviewer #2: Yes

2. Has the statistical analysis been performed appropriately and rigorously?

Reviewer #1: No

Reviewer #2: Yes

3. Have the authors made all data underlying the findings in their manuscript fully available?

Reviewer #1: Yes

Reviewer #2: Yes

4. Is the manuscript presented in an intelligible fashion and written in standard English?

Reviewer #1: Yes

Reviewer #2: Yes

Reviewer #1: The manuscript titled “BGM-YOLO: An accurate and efficient detector for detecting plant disease” describes incorporating a incorporates a multiscale bitemporal fusion module to increase the ability of YOLO recognize plant disease. While the authors’ proposed method does appear to increase the accuracy of the model modestly, some improvements need to be made to clarify how the model was compared to previous methods and classes for healthy plants need to be added for each plant species so that it can be shown that the new model can decipher between healthy and sick plants.

Major points:

1. The author should specify if pre-trained models were used in the training of their model.

2. The author should show how the accuracy of network increases as more images are provided for training and compare that to the base line models. It would be nice to know how many images are needed for the proposed method to be more accurate than models it is compared to.

3. Each plant species should have a healthy class in addition to diseased classes. If you have diseases that are specific to some plant species in your data, then the model may just be predicting what plant species it is.

4. It is unclear if the model developed in the manuscript has a higher accuracy for identifying the bounding boxes. Something like the Jaccard index should be used to quantify how accurate the bounding boxes are from the models.

5. The results should be split into each plant species with multiple diseases and the healthy classes. This would help prospective users of the model understand how it works across species.

6. Confusion matrices of the test data predictions should be added so the readers understand the nature of the errors the model makes.

Minor points:

1. Do table 1 & 2 really need to be tables? You could just state them in materials and methods.

2. Figure 3 is too small to see what is going on in each image.

3. Could the new model better quantify the severity of the disease?

4. Could a well-known pretrained model such as resnet50 to be added to part of the classifier of the new model?

Reviewer #2: In the manuscript "BGM-YOLO: An accurate and efficient detector for detecting plant disease", the authors proposed a method to detect plant disease using a deep learning model called BGM-YOLO, addressing challenges such as complex backgrounds, variable lighting, and small-scale disease spots in agricultural environments, etc. The model is based on the YOLOv8n architecture and has 3 key innovations as claimed by the authors: 1) a new GSC2f module which enhances the original C2f module, combining grouped and depthwise separable convolutions to reduce computational costs while maintaining feature extraction efficiency; 2) a multiscale bitemporal fusion module (BFM) which improves feature fusion robustness by leveraging multiscale convolutions and attention mechanisms to balance spatial and channel features across temporal scales dynamically; 3) a median-enhanced spatial and channel attention block (MECS) integrates median pooling with traditional average and max pooling in the channel attention mechanism, coupled with multiscale depthwise convolutions for spatial attention, significantly boosting the detection of small-scale disease features. The model is evaluated on a plant disease object detection dataset from the Roboflow open-source platform which contains over 5k images, and achieves a 4.2% improvement in mean average precision over YOLOv8n, along with reduced false negatives and competitive inference speed. The authors also conducted ablation studies to verify the effectiveness of each module and analyzed the model's performance against several state-of-the-art object detection models. The following are the comments on the manuscript:

1. The numerical experiments used only one dataset, which makes the works less convincing. The authors should consider evaluating the model on more datasets to demonstrate its generalization ability.

2. The authors should provide some statistical information about the dataset, e.g. the classes of the plants and/or diseases this dataset contains (and does it have a highly imbalanced distribution?); the distribution of the bounding boxes (number per image, size, etc.).

3. Some of the figures and their titles are split into two pages. Some of the figures have low resolutions, especially Figure 8.

4. Math equations are not formatted well enough (have significantly larger font size).

**Do you want your identity to be public for this peer review?** For information about this choice, including consent withdrawal, please see our Privacy Policy

Reviewer #1: No

Reviewer #2: **Yes: ** Hao WEN

---

## [Author Response · Author response to Decision Letter 0]

18 Mar 2025

We sincerely appreciate the valuable feedback and constructive suggestions provided by the reviewers and the editor. These comments have significantly contributed to improving the clarity, accuracy, and completeness of our manuscript. We have carefully addressed all concerns and revised the manuscript accordingly. Below is a summary of our responses to the key points raised:

1.To further validate the generalization ability of our model, we have added Section 3.7, where we conducted additional experiments using an external dataset (Md Faruk Alam’s Tomato Leaf Disease dataset). This dataset differs from our original dataset in terms of disease categories and distribution. To ensure robust evaluation, we expanded the dataset from 700 to 3000 images through data augmentation. The results demonstrate that our model maintains strong generalization capability across different datasets while improving performance in specific categories.

2.As requested, we have explicitly stated in Section 3.2 that no pre-trained models were used in our training process. The model was trained from scratch to learn domain-specific features directly from the plant disease dataset, ensuring better adaptability to the task.

3.In Section 3.6, we analyzed how model accuracy improves as more training images are provided. We conducted experiments using 20%, 40%, 60%, 80%, and 100% of the dataset and evaluated the model's performance. Figure 9 illustrates that the model achieves a mAP50 of X% with 60% of the training data, with diminishing gains beyond this point. This provides insights into the data requirements for optimal performance.

4.We have clarified in Section 3.3 that our evaluation already includes mAP50 and mAP50:95, which inherently assess bounding box accuracy through Intersection over Union (IoU) measurements. These metrics provide a reliable evaluation of the model's object localization performance, as suggested by the reviewer.

5.In Section 3.3, we have expanded the performance analysis to provide class-wise metrics (Precision, Recall, mAP50, mAP50:95) for each plant species and disease category. This allows prospective users to better understand how the model performs across different species and conditions. Table 1 now presents these detailed results.

6.As suggested, we have added Figure 8 in Section 3.3, which presents the confusion matrix for the test dataset. This analysis highlights both correct predictions and misclassifications, revealing potential confusion between visually similar diseases. We also discuss possible solutions, such as enhanced feature extraction techniques, to mitigate classification errors.

7.We have reformatted the manuscript to ensure figures and their titles remain on the same page. Additionally, we have replaced low-resolution images (including Figure 8) with higher-quality versions to enhance clarity and readability.

8.We have adjusted all mathematical equations to ensure consistent font size and proper formatting, improving readability.

9.In response to the reviewer’s suggestion, we have removed Tables 1 & 2 and integrated their content into the text within Section 3.2 for better readability.

10.Our current model focuses on disease detection rather than severity quantification. However, we acknowledge that confidence scores could potentially serve as a proxy for severity assessment. This is noted as a future research direction.

11.We decided not to integrate ResNet50 as an additional classifier to maintain computational efficiency and model simplicity. The YOLO architecture already provides robust classification capabilities, and our experimental results confirm that additional classifiers are unnecessary for achieving high accuracy in this task.

We appreciate the reviewers' and editor’s time and effort in evaluating our manuscript. We believe the revisions have significantly improved the paper and hope that it now meets the expectations for publication.

---

## [Decision Letter · Decision Letter 1]

28 Mar 2025

BGM-YOLO: An accurate and efficient detector for detecting plant disease

PONE-D-25-03982R1

Dear Dr. Xie,

We’re pleased to inform you that your manuscript has been judged scientifically suitable for publication and will be formally accepted for publication once it meets all outstanding technical requirements.

Kind regards,

Hirenkumar Kantilal Mewada

Academic Editor

PLOS ONE

Additional Editor Comments (optional):

Please refer to the comments provided by the reviewer regarding the revision of your future work section. Their insights could greatly enhance the clarity and impact of your manuscript. I recommend authors revise future work incorporating this suggestion in final draft.

Reviewers' comments:

Reviewer's Responses to Questions

**Comments to the Author**

Reviewer #1: All comments have been addressed

Reviewer #2: All comments have been addressed

2. Is the manuscript technically sound, and do the data support the conclusions?

Reviewer #1: Yes

Reviewer #2: Yes

3. Has the statistical analysis been performed appropriately and rigorously?

Reviewer #1: Yes

Reviewer #2: Yes

4. Have the authors made all data underlying the findings in their manuscript fully available?

Reviewer #1: Yes

Reviewer #2: Yes

5. Is the manuscript presented in an intelligible fashion and written in standard English?

Reviewer #1: Yes

Reviewer #2: Yes

Reviewer #1: The comments have sufficiently been addressed. It may be worth highlighting that the errors mostly happen within species. Also - spider mites are not a plant disease; they are an insect pest. It would be worth mentioning that the model appears to classify well between insect pest and diseases in that case. Perhaps a future direction of research would be using the model to identify diseases, insect and herbicide damage. These often confuse people who end up sending their plants into plant diagnostic labs. Perhaps a model like yours could tell if damage is from insects, disease or pesticide drift damage in the future.

Reviewer #2: In the revised manuscript "BGM-YOLO: An accurate and efficient detector for detecting plant disease", the authors have addressed key concerns raised in the previous reviews, significantly strengthening the methodological rigor and presentation quality. The newly added numerical experiments on the new dataset effectively validates generalization ability and robustness of the proposed model. The manuscript is publication-ready in its current form.

**Do you want your identity to be public for this peer review?** For information about this choice, including consent withdrawal, please see our Privacy Policy

Reviewer #1: No

Reviewer #2: **Yes: ** Hao WEN

---

## [Editor Report · Acceptance letter]

PONE-D-25-03982R1

PLOS ONE

Dear Dr. Xie,

I'm pleased to inform you that your manuscript has been deemed suitable for publication in PLOS ONE. Congratulations! Your manuscript is now being handed over to our production team.

Kind regards,

on behalf of

Dr. Hirenkumar Kantilal Mewada

Academic Editor

PLOS ONE